# Research on the Application of Typical Biological Chain for Algal Control in Lake Ecological Restoration—A Case Study of Lianshi Lake in Yongding River

**Pengfei Zhang [1,2], Xiaoyu Cui [2], Huihuang Luo [2,*], Wenqi Peng [2] and Yunxia Gao [1]**

1    School of Municipal and Environmental Engineering, Hebei University of Architecture,
     Zhangjiakou 075000, China; pfzhang1001@163.com (P.Z.); wljgyx@163.com (Y.G.)
2    Department of Water Ecological Environment, China Institute of Water Resources and Hydropower Research,
     Beijing 100038, China; cuixy@iwhr.com (X.C.); pwq@iwhr.com (W.P.)
*    Correspondence: luohh@iwhr.com or luohuihuang@sina.com; Tel.: +86-138-1089-8180

**Abstract:** Maintaining the health of lake ecosystems is an urgent issue. However, eutrophication seriously affects lakes' ecological functions. Eutrophication is also the main target of lake ecological restoration. It is vital to carry out research on lake eutrophication control and energy flow evaluation in ecosystems scientifically. Based on in situ survey results for the aquatic life data for Lianshi Lake from 2018 to 2019, the Ecopath model was used to establish an evaluation index system for the typical biological chain to screen out the key species in the water ecosystem, and the fuzzy comprehensive evaluation (FCE) method was used to screen all the biological chains controlling algae. A combination of the FCE coupled with the Ecopath screening method for typical biological chains for algal control was applied to the Lianshi Lake area; the results show that the typical biological chain for algal control is phytoplankton (Phyt)–zooplankton (Zoop)–macrocrustaceans (Macc)–other piscivorous (OthP). Upon adjusting the biomass of Zoop and Macc in the typical biological chain for algal control to three times that of the current status, the ecological nutrition efficiency of Phyt was increased from 0.308 to 0.906. The material flow into the second trophic level from primary producers increased from 3043 to 8283 t/km$^2$/year. The amount of detritus flowing into primary producers for sedimentation decreased from 7618 to 2378 t/km$^2$/year. Finally, the total primary production/total respiratory volume (TPP/TR) decreased from 9.224 to 3.403, the Finn's cycle index (FCI) increased from 13.6% to 17.5%, and the Finn's average energy flow path length (FCL) increased from 2.854 to 3.410. The results suggest that the problem of eutrophication can be solved by introducing Zoop (an algal predator) and Macc to a large extent, resulting in improved ecosystem maturity. The research results can facilitate decision making for the restoration of urban lake water ecosystems.

**Keywords:** alga control; typical biological chain; Ecopath model; ecological restoration; Lianshi Lake

## 1. Introduction

The rapid expansion of the world's population has exacerbated the degradation of global lake ecosystems [1]. According to statistics, more than 60% of the lakes in the world are in different degrees of eutrophication [2]. The increasing eutrophication of lakes has become a global water environment problem [3] (e.g., for Lake Canyon in the United States [4], Lake Geneva in Switzerland [5], and Sugarloaf Lake in Australia [6]).

The interaction between lake water environments and water ecology is complex. It is important to ensure the integrity of the ecosystem while improving the quality of the water environment [7–9]. Therefore, how to deal with lake eutrophication and restore aquatic ecosystems has become an urgent problem to be solved in current limnology, environmental science, freshwater ecology and other disciplines, and has been closely followed by international scholars [10,11].

At present, the recognized theories of lake ecological restoration mainly include the nutrient salt concentration limit theory by Scheffer [12], multi-steady state theory by Lewontin [13] and biological manipulation theory by Hrbacek [14]. Among them, the biological manipulation theory has the most promising application prospects and has led to many successful cases of lake ecological restoration. For example, the biological manipulation process implemented by putting silver carp (Silc) and bighead carp (Bigc) in Donghu Lake in Wuhan is one of the main reasons for the disappearance of cyanobacteria blooms in the lake [15]. Olin et al. [16] reported that the removal of common carp (Comc) from 10 lakes in southern Finland effectively reduced the biomass of cyanobacteria and the degree of algal outbreaks. Shapiro et al. [17] adjusted the ratio of other piscivarious (Othp) and plankton-eating fish from 1:1.65 to 1:2.2 in the Round Lake, and the concentrations of total nitrogen (TN), total phosphorus (TP) and chlorophyl-a (Chl-*a*) in the lake were reduced to varying degrees. However, biological manipulation cannot enable all ecosystems to achieve the expected ecological functions and may lead to changes in the species diversity of the ecosystem, the decline of the average trophic level in the system, and the destruction of habitats [18,19]. For example, Razlutskij et al. [20], through 72 days of outdoor experiments, found that introducing *Carassius auratus* increased the biomass of Phyt. Recent ecological studies have revealed significant negative effects of crucian carp (Cruc) on the water quality and ecological states of shallow lakes, e.g., increasing nutrient levels, leading to reduced water clarity [21,22]. The current international guidance on the application of biological manipulation technology to the process of lake ecological restoration remains far from sufficient [23].

In light of this, this study intended to establish a screening method for typical biological chains of ecosystems and propose a biomass control strategy for the typical biological chain of alga control. The purpose of this study was to provide strong theoretical support for the management of urban lake eutrophication and water ecological restoration, and it consisted of the following: (1) Key species and target biological chains were screened, based on the Ecopath model; the key species of the ecosystem were analyzed, and the biological chain with phytoplankton (Phyt) as the primary producer and including the key species was selected. (2) Typical biological chains were screened, and an evaluation index that could reflect the efficiency and stability of the biological chain was constructed. The index weight was used to select typical biological chains based on the fuzzy comprehensive evaluation method. (3) The regulatory impact was analyzed, the biomass of key species in the typical biological chain was regulated, and then the impacts on the regulation target and the ecosystem were judged.

## 2. Materials and Methods

### 2.1. Overview of the Study Area and Data Sources

2.1.1. Overview of the Study Area

As a river-type lake in the urban section of the plain of the Yongding River, Lianshi Lake is located at the junction of Mentougou and Fengtai District in the southwest of Shijingshan District (116°6′58″~116°9′34″ E, 39°56′3″~39°53′15″ N), which belongs to the temperate continental monsoon climate, with high temperatures and rain in summer and cold and dry conditions in winter. The total length of Lianshi Lake is 5.8 km, the average width of the lake is 376 m, the widest point is about 500 m, the total water area is 106 hm$^2$, and the average depth of the water body is 1.6 m. The phytoplankton in Lianshi Lake are dominated by *Pseudanabaena moniliformis* (Cyanophinyta) and *Limnothrix*. Lianshi Lake belongs to the northern freshwater city lake. The distribution of the study area and monitoring points are shown in Figure 1.

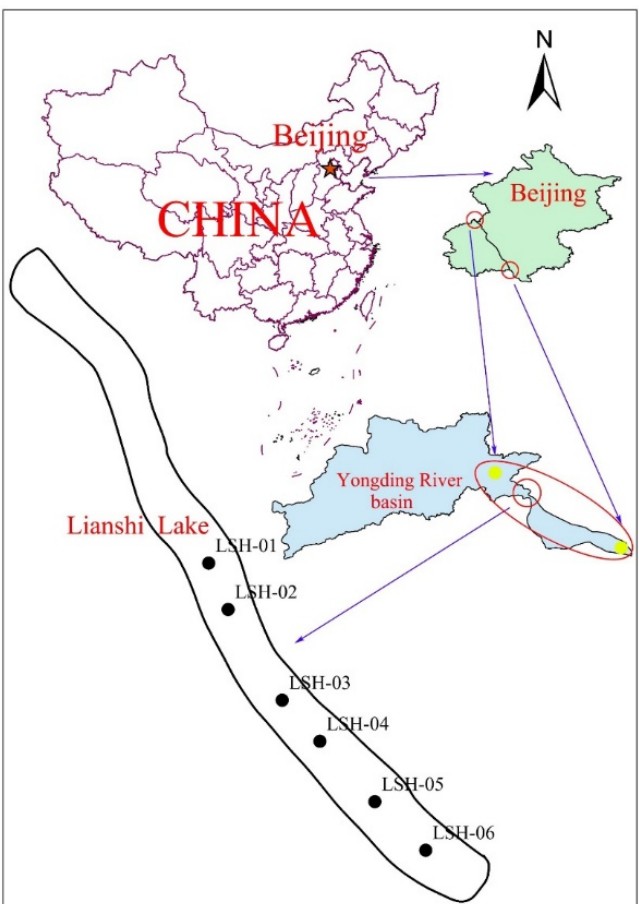

**Figure 1.** The location of Lianshi Lake and the distribution of monitoring points.

2.1.2. Data Sources

The data sources included long-term field sampling survey data, formula calculations and reference comparisons. The main source of the biomass $B_i$ of each function group was a field survey of the biomass at six points in Lianshi Lake from 2018 to 2019. The macroinvertebrates were collected using a Surber sampler with a mesh diameter of 40 mesh (500 μm) and a sampling area of 0.09 m$^2$, and 70% alcohol was added to the test tube bottle to be stored for inspection. The zooplankton were collected using a No. 25 plankton net (200 mesh) and immediately fixed with 4% formaldehyde. For the collection of phytoplankton, a 1 L water sample (0.5 m below the surface of the water body) was collected in a plexiglass water collector, and 15 mL of Lugol's reagent was added for fixation. Sampling areas of 0.5 m × 0.5 m and 1 m × 1 m were used for vascular plants and submerged plants, respectively. In the field, the plant species were distinguished and weighed (wet weight). Fish resources were collected mainly using trawl nets, and the travel distance for each sampling point was no more than 100 m. The biomass of organic detritus was calculated based on the relationship between the primary productivity and water transparency [24] (Equation (1)).

$$logD = 0.954logPP + 0.863logE - 2.41 \tag{1}$$

where $D$ is the detrital biomass (g C·m$^{-2}$), $PP$ is the primary production volume (g C·m$^{-2}$·a$^{-1}$), and $E$ is the average transparency (m).

*2.2. Ecopath Model*

2.2.1. Principle of the Ecopath Model

The Ecopath is an ecological model that can directly determine the structure of an ecological system and describe its energy flow and mass transfer through the principle of nutrition dynamics. The Ecopath model stipulates that each function group (*i*) energy input and output in the ecosystem are balanced. The model uses a set of simultaneous linear equations to define an ecosystem, and each function group is represented by a linear equation [25] (Equation (2)).

$$B_i \times \left(\frac{P}{B}\right)_i \times EE_i = \sum_{j=1}^{n} B_j \times \left(\frac{Q}{B}\right)_i \times DC_{ij} + EX_i \qquad (2)$$

In the formula, $B_i$ is the biomass of the *i*-th function group. $\left(\frac{P}{B}\right)_i$ is the ratio of the annual average production of the *i*-th function group to the annual average biomass, which is the biomass turnover rate. $EE_i$ is the ecological nutrient conversion efficiency of the *i*-th function group and is usually obtained by the calculation of the model [26]. $\left(\frac{Q}{B}\right)_i$ is the ratio of the consumption of the *i*-th function group to the biomass. $DC_{ij}$ is the ratio of the *i*-th the prey group to the total predation of the predator group *j*. $EX_i$ is the output of the *i*-th function group.

2.2.2. Division of Function Groups

The purpose of the establishment of function groups is to merge populations with highly overlapping niches to simplify the food web. Based on the survey results for the Lianshi Lake ecosystem, characteristics of the community, and survival habits of each species, this work identified organisms with similar ecological characteristics, which were grouped into function groups [24]. A total of 14 function groups were set up (Table 1).

**Table 1.** Ecosystem function groups and main types for Lianshi Lake.

| No. | Function Group | Abbreviation for Composition | Included Types |
|-----|----------------|------------------------------|----------------|
| 1 | Other piscivorous | OthP | Horsemouth, yellow catfish |
| 2 | Common carp | Comc | Common carp |
| 3 | Crucian carp | Cruc | Crucian carp |
| 4 | Bighead carp | Bigc | Bighead carp |
| 5 | Silver carp | Silc | Silver carp |
| 6 | Herbivorous fish | HerF | Grass carp, bream |
| 7 | Other fish | OthF | Wheat ear fish, tortoisefish |
| 8 | Macrocrustaceans | Macc | Green prawns, prawns, Chinese mitten crabs, etc. |
| 9 | Other benthos | OthB | *Hydrophilia, Ceratobranchus, Longbrachium, Fanchus, Chironomus,* Chironomidae, etc. |
| 10 | Zooplankton | Zoop | Protozoa, rotifers, cladocerans, copepods, etc. |
| 11 | Phytoplankton | Phyt | Cyanobacteria, green algae, euglena, dinoflagellate, cryptophytes, golden algae, etc. |
| 12 | Submerged macrophytes | SubM | *Potamogeton, Myriophyllum, Hydrilla verticillata, Ceratophyllum,* etc. |
| 13 | Other macrophytes | OthM | Reeds, cattails, etc. |
| 14 | Detritus | Detr | Organic detritus |

2.2.3. Parameter Setting

Some of the parameters refer to lakes with water environmental conditions similar to those of Lianshi Lake, such as the production/biomass (P/B) and consumption/biomass (Q/B) parameters of fish resources that were obtained by querying (Available online: http://www.fishbase.org) (accessed on 20 May 2021). The plant P/B coefficient refers to related research on Taihu Lake [27]. The P/B coefficients of zooplankton (Zoop), other

benthos (Othb) and macrocrustaceans (Macc) were estimated based on the measured data by referring to the research results for Taihu Lake [27], Qiandao Lake [28], Dianshan Lake [29] and Zhushan Bay [30]. The P/Q coefficient refers to recognized data. The values for Zoop, OthB and Macc were 0.05 [31], 0.02 [32] and 0.075 [33], respectively. The food composition ($DC_i$) is shown in Table 2. In addition to ecological investigation and research, we also referred to related research results [27,34], such as those for Zhushan Lake [30] and Qiandao Lake [35,36]. According to the cited literature, the *GS* values for general OthP and herbivorous fish (HerF) were set to 0.2 and 0.41, and the *GS* values for Zoop, OthB and Macc were set as 0.65, 0.94 and 0.7, respectively [31–33,37].

**Table 2.** Food composition data entered into the model.

| No. | Prey/Predator | 1 | 2 | 3 | 4 | 5 | 6 | 7 | 8 | 9 | 10 |
|-----|---------------|-----|-----|-----|-----|-----|-----|-----|-----|-----|-----|
| 1 | OthP | 0.007 | | | | | | | | | |
| 2 | Comc | 0.15 | | | | | | | | | |
| 3 | Cruc | 0.27 | | | | | | | | | |
| 4 | Bigc | | | | | | | | | | |
| 5 | Silc | | | | | | | | | | |
| 6 | HerF | 0.07 | | | | | | 0.006 | | | |
| 7 | OthF | | | | | | | | | | |
| 8 | Macc | 0.38 | | | | | | | | | |
| 9 | OthB | 0.073 | 0.13 | | | | | 0.230 | | | |
| 10 | Zoop | | 0.82 | 0.24 | 0.501 | 0.213 | | 0.620 | 0.350 | 0.005 | 0.009 |
| 11 | Phyt | | 0.048 | | 0.361 | 0.620 | | | 0.300 | 0.022 | 0.801 |
| 12 | SubM | | 0.002 | 0.09 | | | 0.997 | 0.003 | 0.101 | | |
| 13 | OthM | 0.01 | | | | | 0.003 | | | | |
| 14 | Detr | 0.04 | | 0.67 | 0.138 | 0.167 | | 0.141 | 0.350 | 0.872 | 0.190 |
| 15 | Sum | 1 | 1 | 1 | 1 | 1 | 1 | 1 | 1 | 1 | 1 |

### 2.2.4. Model Balance Calculation

The Ecopath model requires the input of six basic parameters: $B_i$, $\left(\frac{P}{B}\right)_i$, $EE_i$, $\left(\frac{Q}{B}\right)_i$, $DC_{ij}$ and $EX_i$. Model balancing was executed, which was established on the basis of conforming to objective laws. The model parameters could be slightly modified to meet the requirements of the model operation, but it was necessary to avoid changing reliable data sources [25]. The nutritional conversion efficiency ($EE_i$) ranged from 0 to 1 [38], the group *EE* was close to 1 in the face of considerable predation pressure, and the underutilized function group had a lower *EE* value. The value range of $GE\left(\frac{P}{Q}\right)$ was generally 0.1–0.3. If the balance test had one or more $EE > 1$, it was necessary to locate which predators caused the problem for specific prey groups in the predation mortality. The level of model confidence is mainly related to the quality and reliability of the acquired data. The accuracy of the Ecopath model was measured using the Pedigree index. The higher the index, the higher the quality of the model. The input and output parameters of the balanced model are shown in Table 3.

**Table 3.** Parameters of Lianshi Lake ecosystem construction.

| Function Group | Biomass (t/km²) | Production/ Biomass | Consumption/Biomass | Eco-Nutrition Efficiency | Production/ Consumption | Proportion of Unassimilated Food |
|----------------|----------------|---------------------|---------------------|--------------------------|-------------------------|----------------------------------|
| OthP | 0.13 | 1.670 | 6.1 | 0.026 | 0.274 | 0.200 |
| Comc | 0.5 | 0.960 | 10.7 | 0.248 | 0.090 | 0.200 |
| Cruc | 0.5 | 1.130 | 12.3 | 0.379 | 0.092 | 0.200 |
| Bigc | 1.8 | 0.990 | 6.9 | 0.001 | 0.143 | 0.200 |
| Silc | 1.2 | 1.100 | 8.0 | 0.001 | 0.138 | 0.200 |

**Table 3.** *Cont.*

| Function Group | Biomass (t/km²) | Production/ Biomass | Consumption/Biomass | Eco-Nutrition Efficiency | Production/ Consumption | Proportion of Unassimilated Food |
|---|---|---|---|---|---|---|
| HerF | 0.27 | 0.987 | 7.1 | 0.778 | 0.139 | 0.410 |
| OthF | 2.3 | 2.155 | 11.0 | 0.001 | 0.196 | 0.410 |
| Macc | 1.58 | 3.090 | 41.0 | 0.062 | 0.075 | 0.700 |
| OthB | 16.141 | 4.130 | 206.5 | 0.099 | 0.020 | 0.940 |
| Zoop | 7.85 | 20.680 | 413.6 | 0.606 | 0.050 | 0.650 |
| Phyt | 47.42 | 185.000 | | 0.308 | | |
| SubM | 1460 | 1.250 | | 0.186 | | |
| OthM | 64 | 1 | | 0.001 | | |
| Detr | 3.230 | | | 0.272 | | |

*2.3. Fuzzy Comprehensive Evaluation Method*

The fuzzy comprehensive evaluation method is based on fuzzy mathematics. There is a certain characteristic of $n$ things to be evaluated; these $n$ things comprise the object set $X = \{x_1, x_2, \ldots, x_n\}$, the factor set $U = \{u_1, u_2, \ldots, u_n\}$ and the evaluation set $V = \{v_1, v_2, \ldots v_m\}$. Suppose that the weight distribution of the factors is the fuzzy subset A on V, denoted as $A = \{a_1, a_2, \ldots, a_n\}$. In the formula, $a_i$ is the weight corresponding to the $i$-th factor $u_i$, and the general rule is $\sum_{i=1}^{n} a_i = 1$.

The evaluation steps for the fuzzy comprehensive evaluation method are as follows:

(1)    Establish a factor set.

Assuming there are a total of $i$ factors of the judged object, the factor universe of the evaluated object $U$ is $U = \{u_1, u_2 \ldots, u_i\}$.

(2)    Determine the membership function.

Assuming that the evaluation level is divided into $i$ levels, the set is $V = \{v_1, v_2, v_3, \ldots v_i\}$.

(3)    Establish a fuzzy relationship evaluation matrix.
(4)    Establish a weight vector.

Determine the weight vector of the judgement factor $A = (a_1, a_2, \ldots, a_n)$. $A$ is the subordination relationship of each factor in $U$ to the thing being judged; it depends on the focus of people when making fuzzy comprehensive judgments, and it is equivalent to assigning weights according to the importance of each factor in the judgment.

(5)    Establish a fuzzy comprehensive evaluation matrix.

According to the calculated maximum membership value, the program selection or category evaluation is performed.

## 3. Results

*3.1. Screening of Typical Biological Chains of Fuzzy Comprehensive Evaluation Coupled with Ecopath*

The Ecopath model was used to screen the key species in the Lianshi Lake ecosystem, identify the types of algae-controlling biological chains containing the key species, and establish a biological chain evaluation index system that could simultaneously express the delivery efficiency and stability. Finally, the fuzzy comprehensive evaluation method was used to screen out the biological chain with the largest weight value.

See Flowchart 2 for specific screening ideas (Figure 2).

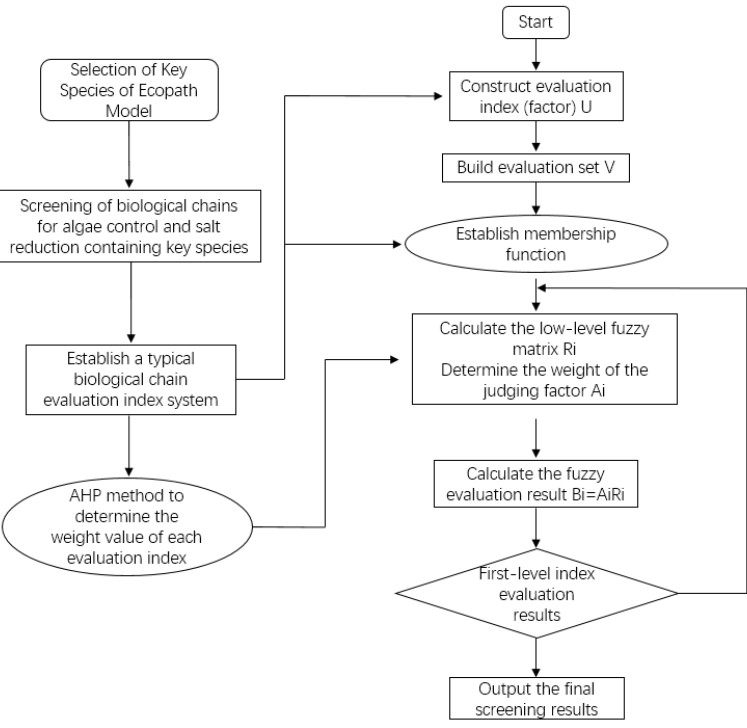

**Figure 2.** Screening flowchart of a typical biological chain.

(1) Analysis of key species in the ecosystem

The key function group plays an important role in the function of the Lianshi Lake ecosystem. This study used the method developed by Libralato et al. [39] to calculate the keystoneness (KS) index of each function group. The one with the largest KS value is regarded as the key function group. Compared to the calculation method for the KS value proposed by Valls et al. [40] and Power et al. [41], this calculation is simpler and comprehensively considers the effects of biomass and capacity. The equation is $KS_i = log[\varepsilon_i(1-p_i)], \varepsilon_i = \sqrt{\sum_{j\neq1}^{n} m_{ij}^2}\varepsilon, p_i = \frac{B_i}{\sum_i^n B_K}$, where $KS_I$ is the criticality index of function group $i$, $\varepsilon_i$ is the total impacts of function group $i$ in the ecosystem (total impacts), and $m_{ij}$ is the value of the mixed nutrition effect of function group $i$ on function group $j$, indicating the mutual relationship between each other's strengths, and $P_i$ is the ratio of the biomass $B_i$ of function group $i$ to the biomass of the entire ecosystem $\sum_k^n B_k$. Since $KS_i$ and $p_i$ are negatively correlated, the criticality index will not be too high due to the high biomass of the function groups (Table 4).

**Table 4.** Criticality data table for each function group of Lianshi Lake.

| No. | Function Group | Biomass | $P_i$ | $\varepsilon_i$ | $KS_i$ | Relative Total Impact |
|-----|----------------|---------|-------|-----------------|--------|------------------------|
| 1 | OthP | 0.13 | 0.0000809 | 0.899516 | −0.04603 | 0.622 |
| 2 | Comc | 0.50 | 0.000311 | 0.147707 | −0.83073 | 0.122 |
| 3 | Cruc | 0.50 | 0.000311 | 0.197666 | −0.7042 | 0.17 |
| 4 | Bigc | 1.80 | 0.00112 | 0.054332 | −1.26543 | 0.0531 |
| 5 | Silc | 1.20 | 0.000747 | 0.017978 | −1.74559 | 0.0175 |
| 6 | HerF | 0.27 | 0.000168 | 0.429709 | −0.3669 | 0.425 |
| 7 | OthF | 2.30 | 0.001431 | 0.845322 | −0.0736 | 0.749 |
| 8 | Macc | 1.58 | 0.000983 | 0.361432 | −0.4424 | 0.328 |
| 9 | OthB | 16.14 | 0.010045 | 1.077714 | 0.028119 | 0.867 |
| 10 | Zoop | 7.85 | 0.004885 | 0.882764 | −0.05628 | 0.701 |
| 11 | Phyt | 47.42 | 0.02951 | 0.914433 | −0.05186 | 0.844 |
| 12 | SubM | 1460 | 0.90857 | 1.014112 | −1.03283 | 1 |
| 13 | OthM | 64 | 0.039828 | 0.007863 | −2.12205 | 0.00724 |

As shown in Figure 3, the first key function groups in the Lianshi Lake ecosystem are submerged macrophytes (SubM). However, other macrophytes (OthM) are redundant in the food web of the entire ecosystem. Other benthos (OthB), Phyt, other fish (OthF) and Zoop have a relative total impact second only to SubM, and their criticality index ranks in the top four.

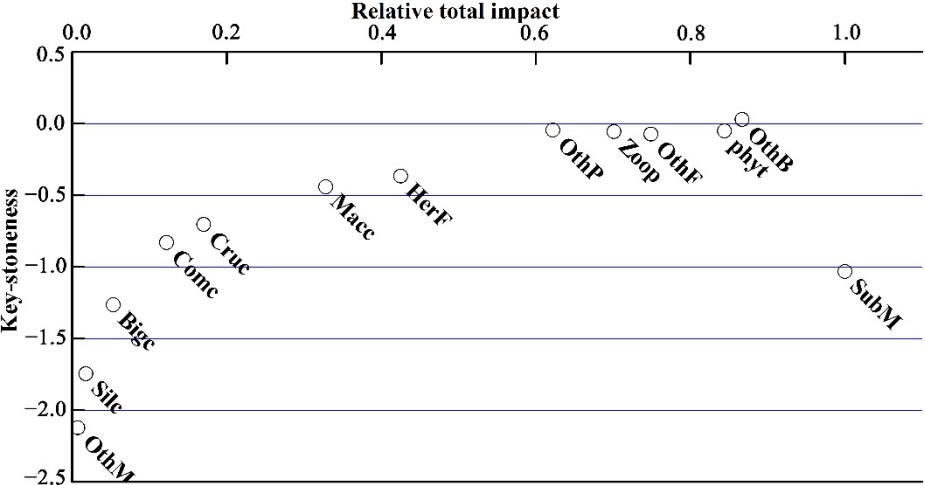

**Figure 3.** Key species in the ecosystem of Lianshi Lake.

(2) Screening of target biological chains for alga control and salt reduction

With the goal of alga control, we identified the biological chain with Phyt and SubM as primary producers, and then determined the biological chain containing key species such as Zoop, OthB and OthF. The results are shown in Figure 4.

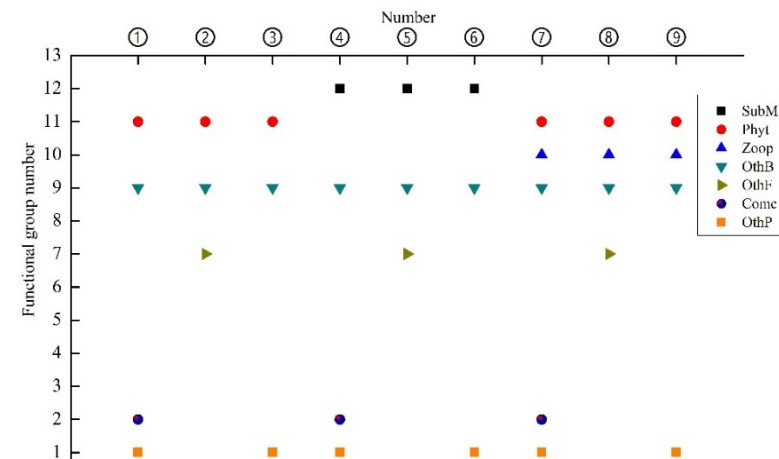

(**a**) Alga control and salt−cutting biological chain containing key species of OthB.

**Figure 4.** *Cont.*

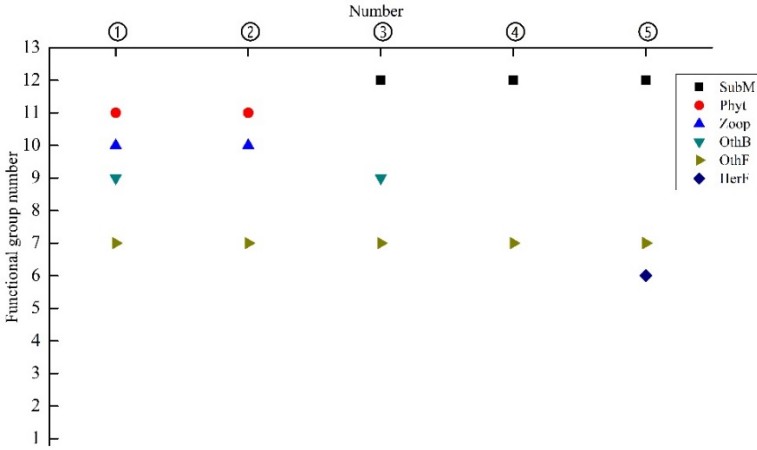

(**b**) Alga control and salt−cutting biological chain containing key species of OthF.

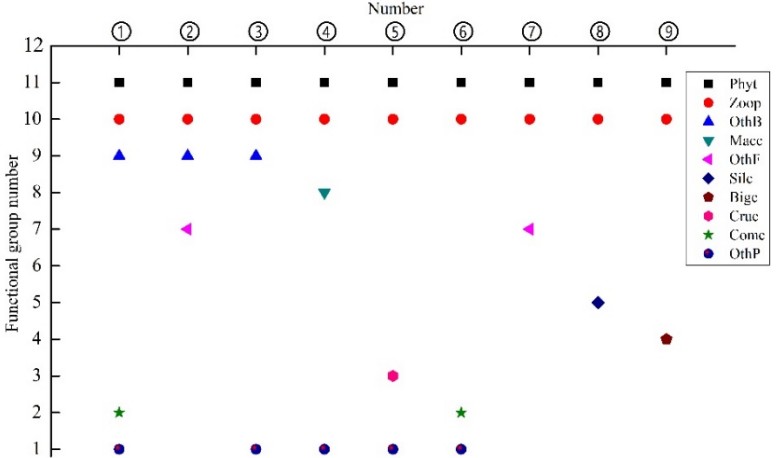

(**c**) Alga control and salt−cutting biological chain containing key species of Zoop.

**Figure 4.** The biological chain of algal control in Lianshi Lake containing key species.

(3) Establishment of evaluation indicators for typical biological chains

In order to express the characteristics of the Lianshi Lake ecosystem, the ecotrophic efficiency (*EE*), production/respiration (*P*/*R*), relative total impact (*RTI*), omnivory index (*OI*) and trophic level (*TL*) are listed as typical indicators for biological chain evaluation (Table 5). The evaluation indicators have the following characteristics: (1) *EE* stands for the utilization and conversion efficiency for the energy contained in the previous trophic organism. The value of EE ranges from 0 to 1, and it is close to 1 in a group with considerable predation pressure. (2) *P*/*R* represents an important indicator of the maturity of the biological chain, which is close to 1 in a mature ecosystem. (3) *RTI* represents the relative total impact of a single function group on the entire ecosystem. (4) *OI* indicates that, when the omnivorous index is zero, the corresponding prey is oriented. (5) *TL* refers to the level of the organism in the food chain of the ecosystem. The higher the trophic level of the organism, the greater the contribution of the food chain to the stability of the ecosystem.

**Table 5.** Basic data of evaluation indicators for each function group of Lianshi Lake.

| Serial Number | Function Group | EE | P/R | OI | TL | RTI |
|:---:|:---:|:---:|:---:|:---:|:---:|:---:|
| 1 | OthP | 0.026 | 0.520 | 0.171 | 3.303 | 0.622 |
| 2 | Comc | 0.248 | 0.126 | 0.048 | 2.958 | 0.122 |
| 3 | Cruc | 0.379 | 0.130 | 0.186 | 2.242 | 0.17 |
| 4 | Bigc | 0.000 | 0.219 | 0.255 | 2.506 | 0.053 |
| 5 | Silc | 0.000 | 0.208 | 0.171 | 2.215 | 0.018 |
| 6 | HerF | 0.778 | 0.308 | 0.000 | 2.000 | 0.425 |
| 7 | OthF | 0.000 | 0.497 | 0.125 | 2.863 | 0.749 |
| 8 | Macc | 0.062 | 0.336 | 0.232 | 2.353 | 0.328 |
| 9 | OthB | 0.099 | 0.500 | 0.005 | 2.005 | 0.867 |
| 10 | Zoop | 0.606 | 0.167 | 0.009 | 2.009 | 0.701 |
| 11 | phyt | 0.308 | - | 0.000 | 1.000 | 0.844 |
| 12 | SubM | 0.186 | - | 0.000 | 1.000 | 1 |
| 13 | OthM | 0.000 | - | 0.000 | 1.000 | 0.007 |
| 14 | Detritus | 0.272 | - | 0.251 | 1.000 | - |

(4) Fuzzy comprehensive evaluation method for screening typical biological chains

The research did not consider the comment-level domain in the fuzzy comprehensive evaluation, and selected the typical biological chain according to the maximum weight (from Equation (3) to Equation (6)):

① The factor domain $U$, $U = (EE, P/R, RTI, OI, TL)$ of the judged object was determined.

② The membership function was established.

The higher the value, the higher the membership function:

$$r_i = \begin{cases} 0 & (x \leq S_1) \\ \frac{x-S_1}{S_2-S_1} & (S_1 < x < S_2) \\ 1 & (x \geq S_2) \end{cases} \tag{3}$$

(I) The determination of the membership functions of $EE$, $P/R$, $RTI$, and $OI$ As the variation range of $EE$, $P/R$, $RTI$, and $OI$ was between 0 and 1, the membership function was:

$$U_A(x) = x \tag{4}$$

(II) The determination of the TL membership function

According to the literature, the lowest trophic level of primary producers in lake ecosystems is 1, and the trophic level of the function group that is generally at the top is 4, so the membership function is:

$$U_A(x) = \frac{x-1}{3} \tag{5}$$

③ Single factor evaluation was performed, and a fuzzy relationship matrix was established.

The fuzzy evaluation matrix was established according to the following formula:

$$R = [r_{ij}] = \begin{bmatrix} r_{11} & \cdots & r_{15} \\ \vdots & \ddots & \vdots \\ r_{51} & \cdots & r_{55} \end{bmatrix} \tag{6}$$

In the formula, in the $i$-th row $R_i = (r_{i1}, r_{i2}, \ldots r_{im})$, $i = 1, \ldots m$, $i$ is the degree of the membership of the $i$-th evaluation factor to the evaluation standards at all levels; for the $j$-th column $R_j = (r_{1j}, r_{2j}, \ldots, r_{nj})$, $j$ is the degree of the membership of each evaluation factor to the $j$-th evaluation standard. Taking the nine biological chains containing the Zoop group as an example, the membership degrees corresponding to different evaluation indicators of different biological chains were calculated and are shown in Table 6 below.

**Table 6.** The membership degrees corresponding to different evaluation indicators of different biological chains with the Zoop group.

| Zoop Group Project Evaluation Index | ① | ② | ③ | ④ | ⑤ | ⑥ | ⑦ | ⑧ | ⑨ |
|---|---|---|---|---|---|---|---|---|---|
| *EE* | 0.257 | 0.338 | 0.260 | 0.251 | 0.330 | 0.297 | 0.457 | 0.457 | 0.457 |
| *P/R* | 0.328 | 0.388 | 0.396 | 0.341 | 0.272 | 0.271 | 0.332 | 0.188 | 0.193 |
| *RTI* | 0.631 | 0.79 | 0.759 | 0.624 | 0.584 | 0.572 | 0.765 | 0.521 | 0.533 |
| *OI* | 0.058 | 0.046 | 0.062 | 0.137 | 0.122 | 0.076 | 0.067 | 0.09 | 0.132 |
| *TL* | 0.418 | 0.323 | 0.360 | 0.389 | 0.380 | 0.439 | 0.319 | 0.247 | 0.279 |

The fuzzy relation matrix with the Zoop group:

$$R = \begin{bmatrix} 0.257 & 0.338 & 0.260 & 0.251 & 0.330 & 0.297 & 0.457 & 0.457 & 0.457 \\ 0.328 & 0.388 & 0.396 & 0.341 & 0.272 & 0.271 & 0.332 & 0.188 & 0.193 \\ 0.631 & 0.790 & 0.759 & 0.624 & 0.584 & 0.572 & 0.765 & 0.521 & 0.533 \\ 0.058 & 0.046 & 0.062 & 0.137 & 0.122 & 0.076 & 0.067 & 0.090 & 0.132 \\ 0.418 & 0.323 & 0.360 & 0.389 & 0.380 & 0.439 & 0.319 & 0.247 & 0.279 \end{bmatrix}$$

④ A weight vector was established.

The weight vector $A = (a_1, a_2, \ldots, a_n)$ of the judgement factor was determined. $A$ is the subordination relationship of each factor in $U$ to the thing being judged. Due to the importance of the distribution of the weight, this study used the analytic hierarchy process to determine the weight of each evaluation index; the hierarchical structure is shown in Figure 5.

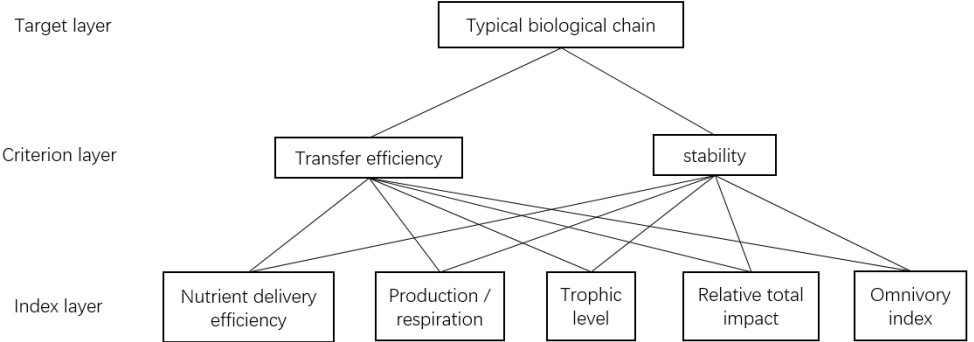

**Figure 5.** Hierarchical structure model.

The judgement matrix $T = \begin{bmatrix} 1 & 4 & 0.33 & 0.143 & 0.25 \\ 0.25 & 1 & 0.25 & 0.143 & 0.2 \\ 3 & 4 & 1 & 0.5 & 3 \\ 7 & 7 & 2 & 1 & 8 \\ 4 & 5 & 0.33 & 0.125 & 1 \end{bmatrix}$. The calculated maximum characteristic value is $\lambda_{max} = 5.42$, the random consensus ratio is $0.0944 < 0.1$, and the consistency test shows that the weight distribution is reasonable, Therefore, the weight of each evaluation index of a typical biological chain is $A = (0.0712, 0.0388, 0.2224, 0.0534, 0.1336)$.

⑤ The comprehensive evaluation results were analyzed.

$$B = AR = (0.2582, 0.2825, 0.2839, 0.2950, 0.2798, 0.2581, 0.2940, 0.2368, 0.2663)$$

It can be seen that the typical biological chain containing the Zoop group as the key species is ④, equal to phytoplankton (Phyt)–zooplankton (Zoop)–macrocrustaceans (Macc)–other piscivorous (OthP).

According to the same calculation principle, the typical algal control biological chain containing OthB and OthF as the key species does not exist.

### 3.2. Analysis of Typical Algal Control Biological Chain and Nutritional Structure and Function Response

Through modeling, it was found that the production of primary producers in the Lianshi Lake ecosystem was 10,661.7 t/km$^2$/year, of which the amount of food consumed was 3043 t/km$^2$/year, and the remaining 71.5% was not consumed by other predators but flowed into the debris, entered recirculation, and became deposited through mineralization. Therefore, the effective removal of plant organisms from the water body plays an important role in controlling the total amount of nutrients, dissolved oxygen and transparency of the water body.

In the Lianshi Lake ecosystem, there are 16 biological chains with Phyt as the primary producer. Among them, the number of biological chains where Phyt is directly grazed by Zoop is the largest; it has nine chains. The transfer efficiency of Phyt is 0.308. The transfer efficiency of Zoop reached 0.606. It can be seen that the types of biological chains that impose significant biological density constraints on algae are mainly Zoop predation on Phyt. Combined with the screening of typical biological chains, a typical biological chain of algal control in the Lianshi Lake ecosystem is phytoplankton (Phyt)–zooplankton (Zoop)–macrocrustaceans (Macc)–other piscivorous (OthP).

This study improved the efficiency of nutrient delivery for primary producers by simulating an increase in biomass of Zoop and Macc. Under the guidance of biological manipulation theory, the key factor in the regulation of the algal control food chain is Zoop. Through the introduction of *Daphnia magna*, the food chain is opened up through insects eating algae and fish-eating insects. A symbiotic system of "*Daphnia magna*–underwater forest–aquatic animals–microbes" was constructed, and the "grass-type clean water state" self-purification system was restored.

The proliferation of ecological capacity continuously increases the biomass of the target species. By observing the changes in other function groups such as the eaten organisms in the model, when the ecological nutrition transfer efficiency of a function group in the model *EE* > 1, the model will become unbalanced; the biomass value of the released species before the imbalance of the model is the ecological capacity. The biomass of Zoop and Macc have been increased to 1.5, 2, 3 and 4 times, respectively. It can be seen from Table 7 that, within the ecological capacity of Zoop and Macc, with increases in the biomasses of both, the ecological nutrition transfer efficiency of Phyt gradually increased, from 0.308 (the current situation) to 0.458 (1.5 times), 0.607 (2 times) and 0.906 (3 times). The flow of material flowing into the second trophic level from primary producers also increased from 3043 (the current situation) to 4353 (1.5 times), 5663 (2 times) and 8283 t/km$^2$/year (3 times).

**Table 7.** Comparison table for overall characteristics of the Lianshi Lake ecosystem.

| Index | Current State | 1.5 Times | 2 Times | 3 Times | 4 Times | Unit |
|---|---|---|---|---|---|---|
| Zoop biomass | 7.85 | 11.775 | 15.7 | 23.55 | 31.4 | t/km$^2$/year |
| Macc biomass | 1.58 | 2.37 | 3.16 | 4.74 | 6.32 | t/km$^2$/year |
| Phyt Eco-nutrition efficiency | 0.308 | 0.458 | 0.607 | 0.906 | 1.204 | - |
| SubM Eco-nutrition efficiency | 0.186 | 0.186 | 0.186 | 0.186 | 0.186 | - |
| The amount of material flowing into the second trophic level from primary producers | 3043 | 4353 | 5663 | 8283 | 10904 | t/km$^2$/year |
| The amount of debris flowing into the primary producer | 7618 | 6308 | 4998 | 2378 | −241.9 | t/km$^2$/year |
| Total primary production/total respiratory volume (TPP/TR) | 9.224 | 6.461 | 4.972 | 3.403 | 2.587 | - |
| Finn's cycle index (FCI) | 13.59% | 14.33% | 15.23% | 17.51% | 20.38% | - |
| Finn's average energy flow path length (FCL) | 2.854 | 2.993 | 3.132 | 3.410 | 3.688 | - |
| Connection coefficient (CI) | 0.225 | 0.226 | 0.226 | 0.226 | 0.226 | - |
| System omnivorous degree (SOI) | 0.092 | 0.092 | 0.093 | 0.093 | 0.094 | - |

The amount of debris flowing into the primary producers for deposition continuously decreased, from 7618 (the current situation) to 6308 (1.5 times), 4998 (two times) and

2378 t/km$^2$/year (three times). The total primary production/total respiratory volume (TPP/TR) continued to decrease from 9.224 (the current situation) to 6.461 (1.5 times), 4.972 (two times) and 3.403 (three times). The Finn's circulation index (FCI) continued to rise from 13.59% (the current situation) to 14.33% (1.5 times), 15.23% (two times) and 17.51% (three times), whereas the Finn's average energy flow path length (FCL) rose from 2.854 (the current situation) to 2.993 (1.5 times), 3.132 (two times) and 3.410 (three times).

Based on the analysis of the nutritional structure of the Lianshi Lake ecosystem, the artificial introduction of Zoop (*Daphnia magna*) and Macc can increase the transfer efficiency and the maturity of the ecosystem. To a certain extent, it can solve the problem of excessive primary production.

## 4. Discussion

### 4.1. Development Characteristics of Lianshi Lake Ecosystem

The total primary production/total respiratory volume (TPP/TR) is an important evaluation index, which is close to 1 in mature ecosystems, far greater than 1 in developing ecosystems, and less than 1 in polluted ecosystems. The Finn's cycling index (FCI) is the ratio of the circulation flow to the total flow in the system, and the Finn's mean path length (FMPL) is the average length of each circulation through the food chain. The higher the ratio of material recycling, the longer the food chain through which the nutrient flows, and the FCI value of a mature ecosystem is close to 1. The current Lianshi Lake TPP/TR is 9.224, the FCI is 13.59%, and the FMPL is 2.854, indicating that the Lianshi Lake ecosystem is immature.

In the typical biological chain of algal control, with an increase in the biomass of Zoop (*Daphnia magna*) and Macc within the ecological capacity, the amount of phytoplankton as a primary producer flowing into the next trophic level gradually increases, the transfer efficiency gradually increases, the FCI and FMPL gradually increase, and the TPP/TR gradually approaches 1, which reduces the risk of lake eutrophication and increases ecosystem maturity to a certain extent.

Therefore, there may be two reasons for the low maturity of the Lianshi Lake ecosystem: first, the biomass of the function group that plays a key role in the Lianshi Lake ecosystem is much lower than the ecological capacity, resulting in insufficient driving force for the ecosystem to develop to a mature state. Second, the biodiversity of Lianshi Lake is low, and the flow of energy to higher levels is hindered. It is recommended to introduce Zoop (*Daphnia magna*) and indigenous herbivorous fishes to build a food chain in order to promote material and energy cycles.

### 4.2. Prospects of Ecopath Model in Lake Ecological Restoration

As early as 1975, Shapiro et al. [17] proposed the biological manipulation theory. Biomanipulation methods have been developed for nearly 50 years; there have been many reports in Western European countries, but this technology has not been widely promoted. This may be because biological manipulation methods involve complex biological networks, and too many factors are affected. Traditional research methods can only study the behavior of individual organisms in simple habitats or competitive environments, and there is very little research on the biological chain that significantly affects the regulation goals and ecosystem conditions. Therefore, it is generally believed that research on complex ecosystems must rely on the guidance of mathematical models or theories [42].

The Ecopath with Ecosim (EWE) model, also known as the ecological channel model, is an ecological model that can assess the true structure of the ecosystem and describe its energy flow and mass balance. The Ecopath model was originally created in 1984 [43]. After years of development, the Ecopath model has become a key tool for ecosystem research. Fuzzy comprehensive evaluation (FCE) is based on fuzzy mathematics [44], which can express the fuzzy relationship between various factors and solve the problem of ambiguity between multi-factor evaluations that cannot be solved by traditional methods, and it has been widely used in the field of policy evaluation and risk assessment. A typical

biological chain of alga control combines the advantages of the Ecopath model and FCE method, and the results show that the biomass of phytoplankton in Lianshi Lake has been effectively controlled and that the ecosystem's maturity has been improved.

Ecopath is a powerful model but is mostly used to assess the condition of the ecosystem and provide scientific management and control solutions. In the future, the Ecopath model will continue to be developed and coupled with other models, such as pollutant diffusion models and ecotoxicology models, which will have important scientific research significance for exploring the restoration of lake ecosystems.

## 5. Conclusions

Based on the survey results for aquatic organisms in Lianshi Lake from 2018 to 2019, this study established, for the first time, a typical biological chain screening method with fuzzy comprehensive evaluation coupled with Ecopath. Among the organisms, Phyt is the primary producer, and the typical biological chain of alga control with Zoop as the key species is phytoplankton (Phyt)–zooplankton (Zoop)–macrocrustaceans (Macc)–other piscivorous (OthP).

In a typical biological chain with significant biological density constraints on algae, when the biomass of Zoop was increased from 7.85 (the current situation) to 23.55 t/km$^2$/year (three times) and Macc was increased from 1.58 (the current situation) to 4.74 t/km$^2$/year (three times), the results show that the ecological nutrition efficiency of Phyt increased from 0.308 (the current situation) to 0.906 (three times), the material flow into the second trophic level from primary producers increased from 3043 (the current situation) to 8283 t/km$^2$/year (three times), the amount of debris flowing into primary producers for sedimentation decreased from 7618 (the current situation) to 2378 t/km$^2$/year (three times), the total primary production/total respiratory volume (TPP/TR) decreased from 9.224 (the current situation) to 3.403 (three times), the Finn's cycle index (FCI) increased from 13.59% (the current situation) to 17.51% (three times), and the Finn's average energy flow path length (FCL) increased from 2.854 (the current situation) to 3.410 (three times). In the typical biological chain of alga control, the artificial release of Zoop (*Daphnia magna*) and Macc can improve the transfer efficiency of phytoplankton as primary producers to a certain extent, reduce the harm caused by eutrophication to lake ecosystems, and improve the maturity of the lake ecosystem.

**Author Contributions:** P.Z. and X.C. designed the research content; P.Z., H.L. and W.P. processed the data and analyzed the results. All the authors participated in the writing of the manuscript. Y.G. and X.C. revised the manuscript. All authors have read and agreed to the published version of the manuscript.

**Funding:** This research was supported by the Major Science and Technology Program for Water Pollution Control and Treatment (No.2018ZX07101005).

**Acknowledgments:** We thank Min Zhang for providing hydrobiological data and Su Wei for assistance with the fieldwork. We would like to thank Guanglei Qiu for his guidance on the translation of the manuscript.

**Conflicts of Interest:** The authors declare no conflict of interest.

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
