# Peer review of "Research on the Application of Typical Biological Chain for Algal Control in Lake Ecological Restoration—A Case Study of Lianshi Lake in Yongding River"

_water, doi:10.3390/w13213079_

Round 1

Reviewer 1 Report

Paper entitled: Research on the Application of Typical Biological Chain of Algae Control in Lake Ecological Restoration—A Case Study of Lianshi Lake in Yongding River by Pengfei Zhang ,Xiaoyu Cui , Huihuang Luo , Wenqi Peng2 and Yunxia Gao is very interesting. Authors made innovative analysis by the Ecopath model to establish an evaluation index system of the typical biological chain to screen out the key species in the water ecosystem. Fuzzy Comprehensive Evaluation method(FCE) is used to screen all the biological chains for controlling algae. I have only minor comments, which I indicated in the pdf file. I accept this paper for publication after minor changes.

Author Response

Dear Editors and Reviewers:

Thank you for your letter and for the reviewers’comments concerning our manuscript entitled “Paper Title” (ID: 1388467). Those comments are all valuable and very helpful for revising and improving our paper, as well as the important guiding significance to our researches. We have studied comments carefully and have made correction which we hope meet with approval. Revised portion are marked in red in the paper. The main corrections in the paper and the responds to the reviewer’s comments are as flowing:

1. Add Author of these theories.

Answer:The authors of these theories have been added to the article.

2. What do the numbers 1-10 refer to in the food composition?

Answer: The number ‘1-12’ in the vertical direction represents the prey, and the number ‘1-10’ in the horizontal direction represents the predator.

3. What do the data in food composition mean?

Answer: The value represents the proportion of predator's food.

4. Specific comments.

Answer:We have made correction according to the Reviewer’s comments. And here we did not list the changes but marked in red in revised paper.

Special thanks to you for your good comments.

Reviewer 2 Report

General comments

The manuscript should be proofread for language. Terminologies should be checked for correctness. For example, the use of “function group” instead of “functional group”. Numerous punctuation errors need to be corrected too.

The weakest part of this work lies in the results and discussion. The authors present numerous results but do not discuss them. This makes it difficult to understand the intended message. Also, some of the equations (if not all) ought to be under the Materials and Methods and not in the results and discussion. I suggest the authors separate the Results from the discussion to give the readers a clear picture of their findings and how this compares to other studies. More so, Algae that form the core of the work is sparingly mentioned throughout the manuscript. Finally, the authors fail to demonstrate how their findings could help in restoring lake ecosystems as mentioned in the title and other places of the manuscript. These need to come out clearly.

Specific comments

Line 14:  …lakes. The…

Line 41: Delete “etc”

Line 69: sought to establish

Line 71: intended to establish

Line 84-91: This whole section needs to be proofread for errors. Some sentences are just hanging.

Line 95-96: What do the authors mean by “formula calculations and reference comparisons” as data sources?

Line 149: “lock” or “look”?

Line 193: Compared to…

Line 205-206: Please rephrase the sentence “However,….”

Line 212-213: Please rephrase

Author Response

Dear Editors and Reviewers:

Thank you for your letter and for the reviewers’comments concerning our manuscript entitled “Paper Title” (ID: 1388467). Those comments are all valuable and very helpful for revising and improving our paper, as well as the important guiding significance to our researches. We have studied comments carefully and have made correction which we hope meet with approval. Revised portion are marked in red in the paper. The main corrections in the paper and the responds to the reviewer’s comments are as flowing:

1. The manuscript should be proofread for language. Terminologies should be checked for correctness. For example, the use of “function group” instead of “functional group”. Numerous punctuation errors need to be corrected too.

Answer:The sentence and punctuation errors in the article have been corrected, and the terminology has been checked in detail.

2. The weakest part of this work lies in the results and discussion. The authors present numerous results but do not discuss them. This makes it difficult to understand the intended message.

Answer:The discussion part of the article has been improved. The discussion part is divided into two aspects. First, we discussed the development characteristics of the Lianshi Lake ecosystem and analyzed the reasons for the low maturity of the Lianshi Lake ecosystem. Second, we discussed the application prospects of Ecopath model in lake ecological restoration. In the future, the Ecopath model will continue to be developed and coupled with other models, such as pollutant diffusion models, ecotoxicology models, etc., which will have important scientific research significance for exploring the restoration of lake ecosystems.

3. some of the equations (if not all) ought to be under the Materials and Methods and not in the results and discussion.

Answer:The equations mentioned in the materials and methods chapter mainly show the basic principles of the method, and some of the equations in the results are the products of the combination of research methods and research purposes. The purpose of the equations in the results is to make readers understand the research ideas clearly.

4. I suggest the authors separate the Results from the discussion to give the readers a clear picture of their findings and how this compares to other studies.

Answer:The results and discussion sections have been separated and improved. Chapter 3 is the result part, and Chapter 4 is the discussion part.

5. Algae that form the core of the work is sparingly mentioned throughout the manuscript.

Answer:We have re-written this part according to the Reviewer’s suggestion. The Phytoplankton in Lianshi Lake are dominated by P.moniliformis of Pseudanabae-na of Cyanophinyta, and Limnothrix.

6. Finally, the authors fail to demonstrate how their findings could help in restoring lake ecosystems as mentioned in the title and other places of the manuscript.

Answer:We have re-written this part according to the Reviewer’s suggestion. First, the typical biological chain of algae control can improve the conversion efficiency of Phytoplankton as primary producers, reduce the biomass of Phytoplankton, and provide theoretical support for solving lake ecosystem problems caused by eutrophication. Second, the study effectively predicts the maturity of lake ecosystems by simulating different calculation conditions.

7. What do the authors mean by “formula calculations and reference comparisons” as data sources?

Answer:The biomass data of the detritus is calculated by the formula. In a similar water environment, the cited parameters are part of the data source.

8. Specific comments

Answer:We have made correction according to the Reviewer’s comments. And here we did not list the changes but marked in red in revised paper.

In all, I found the reviewer’s comments are quite helpful, and I revised my paper point-by-point. Thank you and the review again for your help!

Reviewer 3 Report

Review

Paper title: Research on the Application of Typical Biological Chain of Algae Control in Lake Ecological Restoration–A Case Study of Lianshi Lake in Yongding River.

The authors conducted a modeling study to reveal the most important pathways and study biological chains in Lianshi Lake (China). The results from the Ecopath model as well as the Fuzzy Comprehensive Evaluation method showed the following controlling pathway: Phytoplankton – Zooplankton – Macrocrustaceans – Other piscivorous. The authors modeled different situations and concluded that zooplankton may be used to reduce the level of eutrophication in the lake ecosystem.

All these reasons explain the relevance of the paper by Pengfei Zhang and co-authors submitted to "Water".

General scores.

The data presented by the authors are original and significant. The study is correctly designed and the authors used appropriate methods. In general, the statistical analyses were performed with good technical standards. We authors conducted careful work which may attract the attention of a wide range of specialists focused on ecological modeling and limnology.

Recommendations.

L 96-97. The authors should provide a detailed description of these “field surveys of biomass” including sampling and processing methods. This is important because the authors declared that they used in situ data (L 15).

L 124. The authors used input parameters obtained in other water bodies but provided no reasons for this. They should mention whether or not environmental conditions in the water bodies are similar to explain their selection.

Methods. The authors should include information about the software they used for modeling and analyses.

L 251. “According to literature research”. The authors should provide citations for this research.

L 302. “This study increases biodiversity…”. This statement is unclear. Please, revise.

L 364. “Phyt——Zoop——Macc——OthP.” The authors should use full definitions instead of codes in this section

Specific comments.

L 15. “in-situ” should be italicized.

L 18. Change “Fuzzy Comprehensive Evaluation method” to “The Fuzzy Comprehensive Evaluation method”

L 19. Change “Fuzzy Comprehensive Evaluation method” to “The FCE”

L 20. Change “control. The method was applied” to “control was applied”

L 27. Change “And” to “Finally”

L 32. Change “urban” to “the urban”

L 42. Change “environment” to “environments”

L 55. Change “(2006)” to “[13]” , delete “[13]” at line 57.

L 57. Change “(1984)” to “[14]” , delete “[14]” at line 59.

L 60. Change “changes” to “changes in”

L 62. Change “(2021)” to “[17]” , delete “[17]” at line 64.

L 63. “Carassius  auratus” should be italicized.

L 88. Change “the Lianshi Lake” to “Lianshi Lake”

L 88. Change “376 m. And,” to “376 m and”

L 90. Change “Belongs” to “Lianshi Lake belongs”

L 91. Change “is shown” to “are shown”

L 111. Change “is The” to “is the”

L 121. Change “was grouped” to “were grouped”

L 133. Change “refers” to “refer”

L 142. Please, revise this sentence. It is difficult to understand.

L 173, 175. Change “fuzzy” to “a fuzzy”

L 191. Change “(2006)” to “[36]” , delete “[36]” at line 192.

L 194. Change “(2015)[37] and Power et al.(1996)” to “[37] and Power et al.”

L 212. Change “identified” to “we identified”

L 214. Change “was shown” to “is shown”

L 292. Change “Zoop group” to “the Zoop group”

L 296. Change “preyed” to “grazed”

L 297. Change “the largest” to “largest”

L 300. Change “The  typical” to “a  typical”

L 309. Change “is to continuously increase” to “continuously increases”

L 326. Change “(3 times). And, , the Finn’s” to “(3 times) whereas Finn’s”

L 330. Change “biodiversity, increase” to “biodiversity and”

L 331. Change “excess” to “excessive”

L 333. Change “Lianshi Lake ecosystem” to “the Lianshi Lake ecosystem”

L 334. Change “As early as 1975, Shapiro et al.(1975) proposed biological manipulation theory[14]” to “As early as in 1975, Shapiro et al. [14] proposed biological manipulation theory”

L 350. Change “Typical” to “A typical”

L 364. Change “Pops” to “POPs”

L 375. Change “the Finn’s” to “Finn’s”

L 423, 434. “Carassius carassius” should be italicized.

L 428. “Carassius gibelio” should be italicized.

Author Response

Dear Editors and Reviewers:

Thank you for your letter and for the reviewers’comments concerning our manuscript entitled “Paper Title” (ID: 1388467). Those comments are all valuable and very helpful for revising and improving our paper, as well as the important guiding significance to our researches. We have studied comments carefully and have made correction which we hope meet with approval. Revised portion are marked in red in the paper. The main corrections in the paper and the responds to the reviewer’s comments are as flowing:

1. L 96-97. The authors should provide a detailed description of these “field surveys of biomass” including sampling and processing methods. This is important because the authors declared that they used in situ data (L 15).

Answer:We have re-written this part according to the Reviewer’s suggestion. The macroinvertebrate were collected using a Surber sampler with a mesh diameter of 40 mesh (500μm) and a sampling area of 0.09m2, and 70% alcohol was added to the test tube bottle to be stored for inspection. The zooplankton was collected using No. 25 plankton net (200 mesh) and fixed on site with 4% formaldehyde solution.For the col-lection of phytoplankton, a 1L water sample (0.5m below the surface of the water body) was collected in a plexiglass water collector, and 15 mL of Lugol's reagent was added for fixation. The collection of Vascular plants used a sampling area of 1m×1m, and submerged plants used a sampling area of 0.5m×0.5m , then the plant species are dis-tinguished and weighed (wet weight). Fish resources were collected mainly through trawl nets, and the travel distance of each sampling point was not more than 100 meters.

2. L 124. The authors used input parameters obtained in other water bodies but provided no reasons for this. They should mention whether or not environmental conditions in the water bodies are similar to explain their selection.

Answer:This issue is within my consideration. The input parameters used for reference in the Ecopath model all come from lakes with similar water environmental conditions. To a certain extent, it can guide the construction of Ecopath model. The corresponding part has been improved in the article.

3. The authors should include information about the software they used for modeling and analyses.

Answer:The information of the model software has been introduced in the article.

4. L 251. “According to literature research”. The authors should provide citations for this research.

Answer:References have been added to the corresponding positions in the article.

5. L 302. “This study increases biodiversity…”. This statement is unclear. Please, revise.

Answer:We are very sorry for our negligence of “This study increases biodiversity…”. “This study increases biodiversity…”has been changed to “This study improves the efficiency of nutrient delivery for primary producers by simulating the increase in biomass of Zoop and Macc. ”

6. L 364. “Phyt——Zoop——Macc——OthP.” The authors should use full definitions instead of codes in this section.

Answer:"Phyt——Zoop——Macc——OthP." has been changed to "Phytoplankton(Phyt)—Zooplankton(Zoop)—Macrocrustaceans(Macc)—Other pisciv-orous (OthP)."

7. Specific comments.

Answer:We have made correction according to the Reviewer’s comments. And here we did not list the changes but marked in red in revised paper.

In all, I found the reviewer’s comments are quite helpful, and I revised my paper point-by-point. Thank you and the review again for your help!

Round 2

Reviewer 2 Report

Please proofread for minor language errors

Author Response

Dear Editors and Reviewers:

Thank you for your letter and for the reviewers’comments concerning our manuscript entitled “Paper Title” (ID: 1388467). Those comments are all valuable and very helpful for revising and improving our paper, as well as the important guiding significance to our researches. We have studied comments carefully and have made correction which we hope meet with approval. Revised portion are marked in red in the paper. The main corrections in the paper and the responds to the reviewer’s comments are as flowing:

  1. Please proofread for minor language errors

Answer:We have made correction according to the Reviewer’s comments. And here we did not list the changes but marked in red in revised paper.

Reviewer 3 Report

Second Review

Paper title: Research on the application of typical biological chain of algae control in lake ecological restoration—a case study of Lianshi Lake in Yongding River.

The authors have improved the paper and provided sufficient responses to my comments.

A few minor text revisions are still required:

L 91. Change “P.moniliformis  of  Pseudanabaena  of  Cyanophinyta,  and  Limnothrix.” to “Pseudanabaena  moniliformis  (Cyanophinyta)  and  Limnothrix.” “Pseudanabaena  moniliformis”  and “Limnothrix” should be italicized.

L 99. Change “macroinvertebrate” to “macroinvertebrates”

L 103. Change “fixed on site” to “fixed immediately”

L 105. Change “The collection of Vascular plants used a sampling area of 1m×1m, and submerged plants used a sampling area of 0.5m×0.5m , then the plant species are distinguished and weighed (wet weight).” to “Sampling areas of 0.5m×0.5m and 1m×1m were used for Vascular plants and submerged plants, respectively. In the field, the plant species were distinguished and weighed (wet weight).”

L 140. Change “Among other biological parameters, Some parameters referred to lakes with similar  water environmental conditions as Lianshi Lake. the production/biomass (P/B) and con-

sumption/biomass  (Q/B)  parameters  of  fish  resources” to “Some parameters referred to lakes with similar water environmental conditions as Lianshi Lake such as the production/biomass (P/B) and consumption/biomass  (Q/B)  parameters  of  fish  resources”

L 325. Change “is to continuously increase” to “continuously increases”

Author Response

Dear Editors and Reviewers:

Thank you for your letter and for the reviewers’comments concerning our manuscript entitled “Paper Title” (ID: 1388467). Those comments are all valuable and very helpful for revising and improving our paper, as well as the important guiding significance to our researches. We have studied comments carefully and have made correction which we hope meet with approval. Revised portion are marked in red in the paper. The main corrections in the paper and the responds to the reviewer’s comments are as flowing:

  1. Specific comments.

Answer:We have made correction according to the Reviewer’s comments. And here we did not list the changes but marked in red in revised paper.

In all, I found the reviewer’s comments are quite helpful, and I revised my paper point-by-point. Thank you and the review again for your help!
